# Preferential inhibition of adaptive immune system dynamics by glucocorticoids in patients after acute surgical trauma

Edward A. Ganio[1,6], Natalie Stanley[1,6], Viktoria Lindberg-Larsen [2,6], Jakob Einhaus [1], Amy S. Tsai [1], Franck Verdonk [1], Anthony Culos [1], Sajjad Gahemi[1,3], Kristen K. Rumer[1], Ina A. Stelzer [1], Dyani Gaudilliere[4], Eileen Tsai[1], Ramin Fallahzadeh[1], Benjamin Choisy[1], Henrik Kehlet [5], Nima Aghaeepour [1,6], Martin S. Angst [1,6] & Brice Gaudilliere [1,6✉]

Glucocorticoids (GC) are a controversial yet commonly used intervention in the clinical management of acute inflammatory conditions, including sepsis or traumatic injury. In the context of major trauma such as surgery, concerns have been raised regarding adverse effects from GC, thereby necessitating a better understanding of how GCs modulate the immune response. Here we report the results of a randomized controlled trial (NCT02542592) in which we employ a high-dimensional mass cytometry approach to characterize innate and adaptive cell signaling dynamics after a major surgery (primary outcome) in patients treated with placebo or methylprednisolone (MP). A robust, unsupervised bootstrap clustering of immune cell subsets coupled with random forest analysis shows profound (AUC = 0.92, p-value = 3.16E-8) MP-induced alterations of immune cell signaling trajectories, particularly in the adaptive compartments. By contrast, key innate signaling responses previously associated with pain and functional recovery after surgery, including STAT3 and CREB phosphorylation, are not affected by MP. These results imply cell-specific and pathway-specific effects of GCs, and also prompt future studies to examine GCs' effects on clinical outcomes likely dependent on functional adaptive immune responses.

[1] Department of Anesthesiology, Perioperative and Pain Medicine, School of Medicine, Stanford University, Stanford, CA, USA. [2] The Lundbeck Foundation Center for Fast-track Hip and Knee replacement, Copenhagen, Denmark. [3] Digital Technologies Research Centre, National Research Council Canada, Toronto, ON, Canada. [4] Division of Plastic and Reconstructive Surgery, Department of Surgery, School of Medicine, Stanford University, Stanford, CA, USA. [5] Section of Surgical Pathophysiology 7621, Rigshospitalet, Blegdamsvej 9, DK-2100 Copenhagen, Denmark. [6] These authors contributed equally: Edward A. Ganio, Natalie Stanley, Viktoria Lindberg-Larsen, Nima Aghaeepour, Martin S. Angst, Brice Gaudilliere. ✉email: gbrice@stanford.edu

The use of glucocorticoids (GCs) for the management of acute inflammatory conditions including sepsis and traumatic injury remains controversial[1,2]. Despite equivocal results regarding their beneficial or potentially harmful effects, GCs are frequently administered to patients undergoing major surgery[3–7]. GCs effectively decrease the incidence of postoperative nausea and vomiting[8]—a process predominantly driven by direct central effects[9]—and some studies suggest additional benefits including attenuation of postoperative pain and fatigue[10,11]. However, such findings are inconsistent. Concerns regarding the use of GCs in acutely injured patients include increased infection risk and impaired wound and bone healing. As a result, administration of GCs in bone fusion surgery is frequently discouraged out of concern for impaired functional outcomes[12]. In addition, a scarcely explored but highly relevant question in cancer surgery is whether GCs promote microseeding and, consequently, metastatic disease[13]. Considering the widespread administration of GCs in surgical patients, an in-depth analysis of the effects of GCs on the immune response to surgical injury is critical to establish a biological basis that can guide their safe and effective clinical use.

While the immune modulating properties of GCs have been examined in various clinical contexts, there is sparse information regarding their effects in patients who suffer from acute traumatic injury[14]. Studies examining the effects of GCs on the production of circulating inflammatory mediators[15] or on the distribution patterns of select immune cell subsets[16] after surgery have provided important insights. Specifically, prior studies have shown that GCs differentially and dose-dependently attenuate the production of both pro-inflammatory (e.g., IL-6)[17,18] and anti-inflammatory circulating cytokines (e.g., IL-10)[17–19]. However, the binding of GCs to the glucocorticoid receptor (GR) results in a pleiotropic modulation of multiple inflammatory signaling pathways in both innate and adaptive immune cells[14]. As such, single-cell techniques capturing functional responses in all major immunological compartments will provide much needed insight as to how GCs modulate the complex immune response to acute injury[20].

Our group has recently utilized mass cytometry—a high-dimensional, single-cell flow cytometry technology[21,22]—to comprehensively interrogate the peripheral immune system in patients undergoing total hip arthroplasty (THA) surgery[23,24]. THA is a compelling model to study the effect of GCs on the human immune response to major trauma as it is associated with significant tissue injury, which initiates a stereotypical and coordinated immune response: within hours of injury, neutrophils, monocytes (MCs), and natural killer (NK) cells, activated by alarmins and inflammatory cytokines, are recruited to the site of injury, while compensatory immunosuppressive events, including decreased T cell frequencies and effector functions, evolve in parallel[25,26]. As innate and adaptive immune cells respond to multiple environmental cues, they integrate complex extracellular signals into coordinated signaling responses that enable wound healing and recovery[23,24,27–34]. Differences in response patterns can impact clinical recovery trajectories[35]. For example, our group has recently shown that accentuated signaling responses in innate immune cells, including elements of the JAK/STAT and MAPK/CREB signaling pathways in classical monocytes (cMCs), strongly correlated with delayed resolution of pain and prolonged functional impairment of the operated joint[23,24].

Here, we apply a high-dimensional mass cytometry assay to characterize the immune-modulating effects of a single dose of 125 mg methylprednisolone (MP) in a randomized control trial of patients undergoing THA (NCT02542592). Our primary aim is to provide an in-depth profile of peripheral immune cell distribution and intracellular signaling responses, thereby building a high-resolution cell atlas of immune system dynamics after surgery in patients treated with placebo or GCs (primary outcome). Our secondary aim is to examine whether single-dose administration of GCs would improve patient-centered recovery outcomes, including pain and function (secondary outcomes), which were previously predicted by specific immune response patterns after surgical injury[23,24]. Our data indicate that GCs profoundly alter adaptive immune cell signaling responses after surgery while sparing key innate cell signaling responses previously associated with surgical recovery from pain and functional impairment. Observed cell-specific immune modulation by GCs is consistent with the fact that GC did not alter measured parameters of surgical recovery. The results provide the basis for future studies examining the effect of perioperative GC administration on surgical outcomes that may be particularly affected by adaptive immune cell alterations.

## Results

**Patient and clinical characteristics.** Sixty-three patients undergoing primary THA were randomized to receive a single preoperative 125 mg dose of intravenous MP (MP group) or isotonic saline (control group) on the day of surgery. Patient characteristics have been described in prior analyses focused on the effect of GCs on cardiovascular regulation[36] (NCT02445898) and glucose homeostasis[37] (NCT02332603), and are summarized in Table 1. Samples suitable for analysis with mass cytometry were available for 58 patients, 30 randomized to the control group and 28 randomized to the MP group (Supplementary Fig. 1).

**Assessment of immunological trajectories after surgery.** Serial whole blood samples collected at baseline (pre-surgery) and at 1, 6, 24, 48 hours (h), and 2 weeks (wk) after surgery were analyzed using a 47-parameter mass cytometry immuno-assay (Fig. 1). Twenty-six cell phenotype markers were simultaneously assessed to characterize major innate and adaptive immune cell subsets (Supplementary Fig. 2). In addition, the activity of 11 intracellular proteins that are activated after surgery, including elements of the MAPK, CREB, NF-κB and JAK/STAT signaling pathways were quantified on a per-cell basis. Intracellular markers were chosen based on our previous reports showing that variation in the activity pattern of these signaling responses correlated with sentinel clinical recovery parameters, including the resolution of pain and the function of the operated hip[23,24].

**Table 1 Patient and procedural characteristics.**

| Demographics | Control (received placebo, $n = 30$) | Treatment (received MP, $n = 28$) |
|---|---|---|
| Age, years, mean (SD) | 67.2 (6.7) | 67.3 (5.4) |
| Gender, n (% female) | 18 (60%) female | 12 (43%) female |
| Race, n (%) | | |
| White | 29 (97) | 28 (100) |
| Other | 1 (3) | 0 (0) |
| BMI, mean (SD) | 27.4 (4.3) | 26.9 (4.1) |
| **Surgery** | | |
| Duration, min, mean (SD) | 51.2 (14.1) | 61.1 (18.4) |
| Blood loss, mL, mean (SD) | 238.7 (148.0) | 299.6 (166.0) |
| Intraoperative fluids, mL, mean (SD) | 868.3 (200.2) | 971.4 (279.7) |
| **Anesthesia** | | |
| Propofol sedation dose, mg, mean (SD) | 165.5 (137.3) | 178.5 (129.7) |
| Bupivacaine dose, mg mean (SD) | 14.8 (1.2) | 15.0 (1.01) |

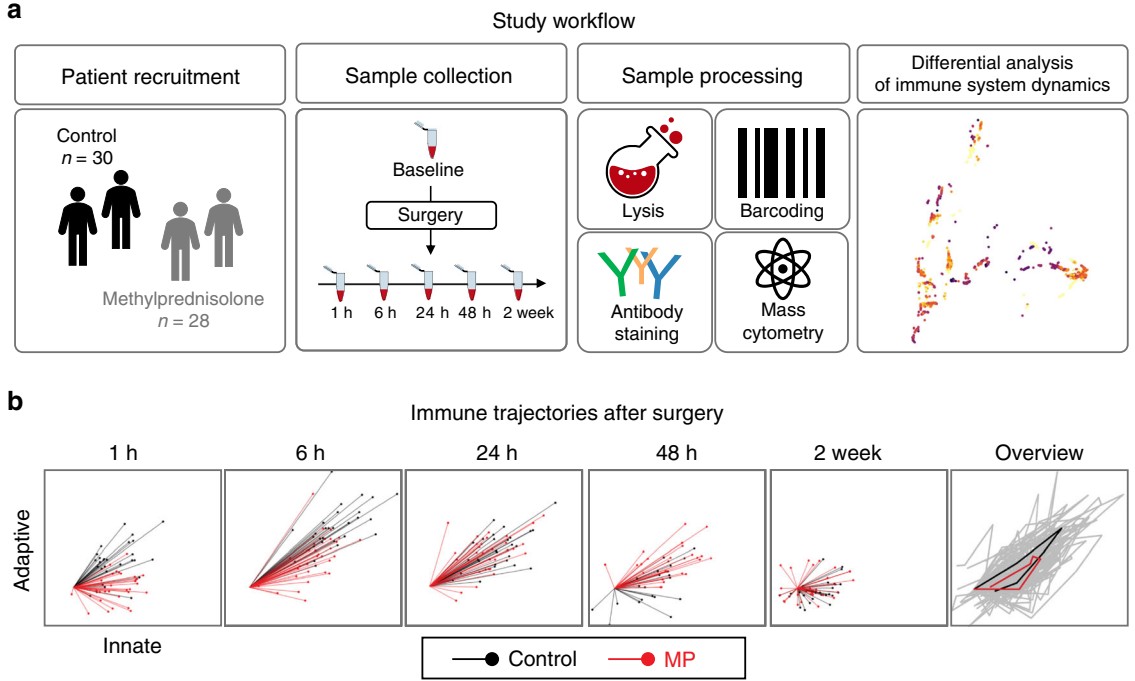

**Fig. 1 Study workflow. a** In a double-blind study, patients were randomized to receive a single preoperative dose of 125 mg methylprednisolone (MP, n = 28 patients) or saline placebo (n = 30 patients). Peripheral blood and clinical outcomes data were collected prior to surgery (baseline) and at the indicated time points after surgery. After erythrocyte lysis, peripheral immune cells were barcoded, stained with cell-phenotyping and intracellular cell-signaling antibodies, and analyzed by mass cytometry. Unsupervised bootstrapped clustering of immune cell subsets followed by random forest analysis was performed to identify differential immune cell dynamics in MP vs. control groups. **b** A non-linear dimensional reduction algorithm (Isomap) showing individual patients' immunological trajectories after surgery along the innate (X) and adaptive (Y) axes (MP in red, control in black). Snapshots are shown for the 1 h, 6 h, 24 h, 48 h and 2 wk time points (animated trajectories can be found in Supplementary Movie 1). Right panel. Overview of median trajectories for the MP and control groups are shown in red and black.

Twenty-one innate and adaptive immune cell subsets were manually gated using an established strategy[38]. The resulting immunological dataset was divided into two sets of immune features, quantifying the distribution and intracellular signaling activity for 9 innate (first set) and 12 adaptive (second set) immune cell subsets. A non-linear dimensionality reduction algorithm (Isomap) was employed to dynamically plot surgery-evoked immune trajectories along the innate and adaptive axes over the two-week postoperative course (Supplementary Fig. 3, Supplementary Movie 1). Immunological trajectories evolved along the innate and adaptive axes immediately after surgery. This suggests that elements of both the innate and adaptive compartments are mobilized early and jointly after traumatic injury. Our results are consistent with recent transcriptomic and flow cytometry analyses[23,34] and challenge the traditional view of sequential engagement of innate and adaptive compartments after traumatic injury.

**MP modulates immune responses for at least 48 h after surgery**. Separate inspection of the immunological trajectories for patients randomized to MP or placebo treatment (Supplementary Fig. 3, Supplementary Movie 1) revealed pronounced differences between the two groups for at least 48 h after surgery. However, trajectories diverged more along the adaptive (Y) than the innate (X) axis, indicating that MP affected the adaptive immune compartment more prominently than the innate compartment.

To complement the Isomap analysis with a quantitative and cell-specific evaluation, an unsupervised clustering algorithm was applied to the mass cytometry dataset (excluding neutrophils which were analyzed separately). This algorithm was developed to compute statistics quantifying differences between patient groups,

while enabling the visualization of group-level statistics on a per-cell cluster basis for a systems-level mapping of clinically relevant and cell subset specific differences between patient groups (see "Methods"). Cells were clustered into coherent subpopulations based on the expression of all cell phenotype markers using a robust bootstrapped meta-clustering algorithm. All clusters were projected onto a two-dimensional cell atlas for visual interpretation using principal component analysis (PCA) (Fig. 2a). Major innate and adaptive cell populations were identified and contoured based on the expression of canonical surface markers. Differences in cell frequency and intracellular signaling responses between the control and MP treatment groups were quantified in each cell cluster at each time point and visualized in the two-dimensional PCA plot (Fig. 2b).

A random forest (RF) algorithm was applied to the dataset comprised of all cell cluster features (including frequency and signaling features) to estimate the magnitude of differences for features separating the placebo group from the MP group at each time point (Fig. 2c). Using a leave-group-out cross validation procedure (see methods) the RF model predicted the probability that each sample belonged to the MP group. The control and MP groups differed significantly at 1 h (AUC = 0.91, $p = 1.03E{-}7$), 6 h (AUC = 0.92, $p = 3.16E{-}8$), 24 h (AUC = 0.85, $p = 3.81E{-}6$), and 48 h (AUC = 0.76, $p = 2.3E{-}3$) after surgery, but not at baseline (AUC = 0.52, $p = 0.76$), or at 2 wk (AUC = 0.48, $p = 0.84$) after surgery. These results indicate that a single dose of MP result in a wide-spread and cell-specific modulation of the immune response to surgery for at least 48 h after surgery. This is consistent with the time course of non-specific alterations of plasma inflammatory markers including C-reactive protein as previously reported for this cohort[39].

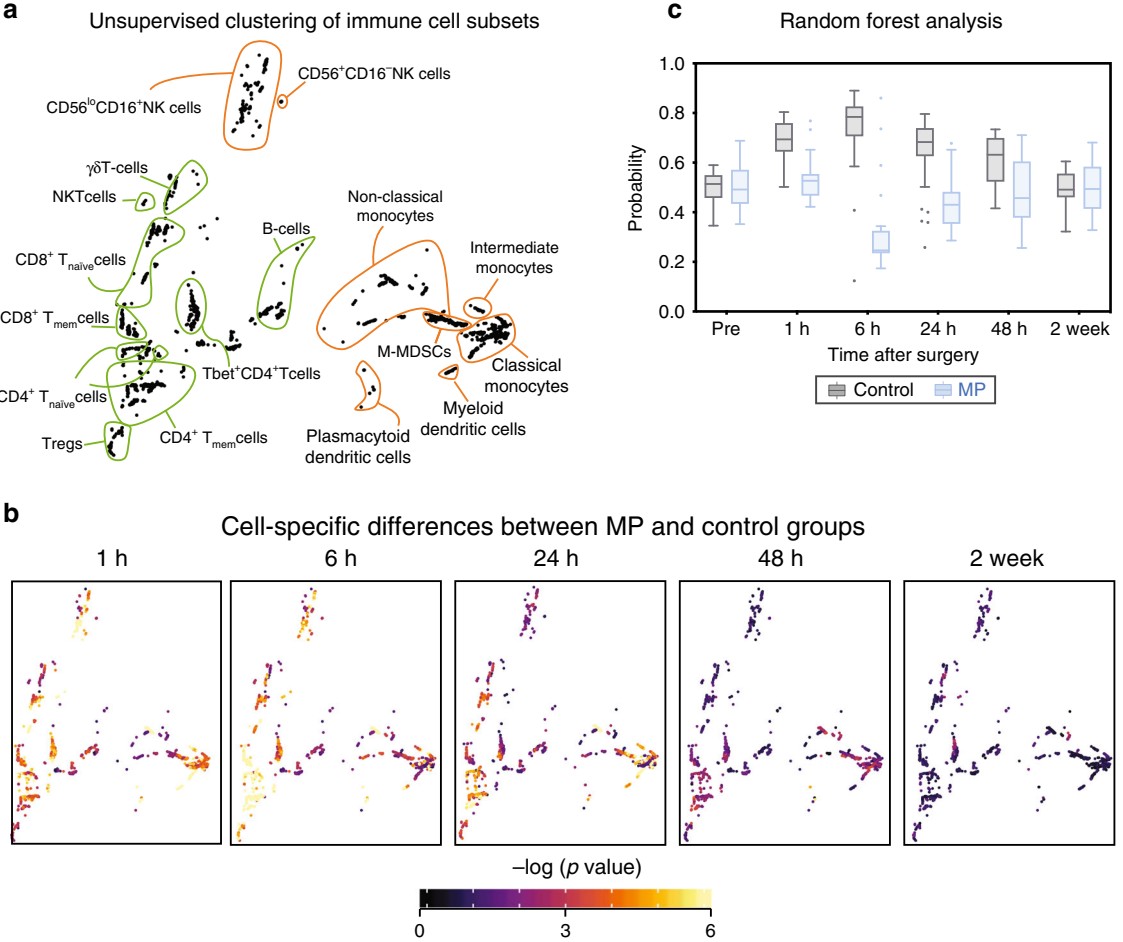

**Fig. 2 A high-resolution atlas detailing peripheral immune cell alterations by MP after surgery. a** Immune cells were clustered based on the expression of all phenotypic markers using an unsupervised bootstrapped clustering algorithm. The clusters were projected into two dimensions and major immune cell compartments were identified based on phenotypic marker expression (contoured in orange/green for innate/adaptive immune compartments, respectively). **b** Univariate *p*-values (two-sided Wilcoxon Rank Sum Test) were computed for each cluster at each time point to quantify the difference in functional marker expression or cell frequency between samples in the control ($n = 30$ patients) and MP ($n = 28$ patients) groups. At each time point, clusters were colored by the best univariate p-value observed for cell frequency and functional marker expression. **c** A Random Forest model was trained to classify patients in the control or MP group at each timepoint based on cluster-derived cell frequency and intracellular signaling responses. The boxplot depicts the probability predicted by the Random Forest model that samples from patients in the control (gray) or MP (blue) group were allocated to the MP group. The model revealed that samples from placebo- or MP-treated patients were distinguishable at 1 h (AUC = 0.91, $p = 1.03E-7$), 6 h (AUC = 0.92, $p = 3.16E-8$), 24 h (AUC = 0.85, $p = 3.81E-6$), and 48 h (AUC = 0.76, $p = 3.2E-3$) after surgery (two-sided Wilcoxon rank-sum test, p-values calculated for each unique model). All boxplots show median values, interquartile range, whiskers of 1.5 times interquartile range.

The RF analysis, anchored in a statistically stringent cross-validation method, indicated that the effect of MP was most pronounced for the 1, 6, and 24 h time points (Fig. 2c). These perioperative time points were examined in more detail to determine the cell- and signaling-specific effects of MP (Supplementary Fig. 3). Cell frequencies and signaling responses were also quantified in manually gated immune cell subsets (Supplementary Fig. 4) to corroborate observations contained in the immune atlas using a univariate statistical approach (Wilcoxon Rank Sum Test).

**MP reorganizes the phenotypic immune landscape after surgery.** We examined the effect of MP on the frequency of immune cell subsets at 1, 6, and 24 h after surgery (Fig. 3a). MP altered the frequencies of several immune cell subsets, most prominently at 1 and 6 h after surgery. In the adaptive compartment, cell frequencies in CD4$^+_{naive}$ and CD4$^+_{mem}$ T cell subsets (including Tbet$^+$CD4$^+$ T cell subsets) decreased at 1 and 6 h after surgery in the MP group compared to the control group (Fig. 3a, b). Little or

no difference was detected for the frequencies of B cell or CD8$^+$ T cell subsets. In the innate compartment, cMCs, ncMCs, and M-MDSC frequencies decreased, while the frequency of a subset of CD56$^{lo}$CD16$^+$ NK cells and neutrophils increased at 1 and 6 h after surgery (Fig. 3a, c). The frequency of mDCs decreased at 1 and 6 h, but increased at 24 h after surgery.

MP can affect immune cell frequencies through several mechanisms including alteration of intracellular signaling responses implicated in the mobilization, adhesion, proliferation and survival of these cells in the peripheral immune compartment[6]. In addition, insight into the effects of MP on cell type-specific signaling responses to surgery may be more instructive than considering immune cell frequencies in isolation, as prior studies suggest functional attributes are powerful correlates of surgical outcomes[23,29]. We therefore evaluated the effect of MP on intracellular signaling dynamics.

**MP reorganizes the functional immune landscape after surgery.** The median intracellular signaling activity was quantified for each

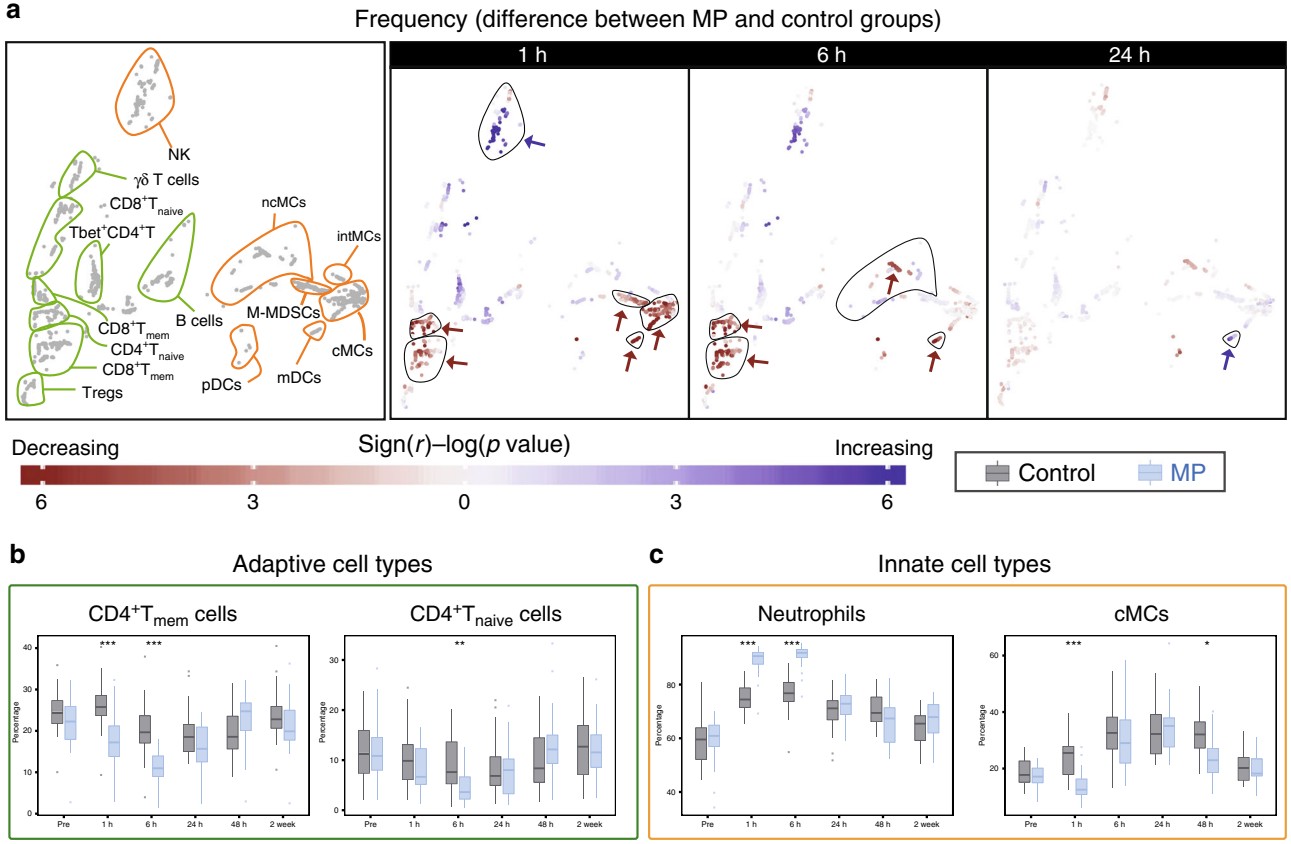

**Fig. 3 Alteration of innate and adaptive immune cell frequencies by MP. a** Immune cell atlas depicting differences in cell frequency between the MP and control groups at 1, 6 and 24 h after surgery (expressed as % of CD45$^+$ mononuclear cells, with the exception of neutrophils, which are expressed as % of total live cells). Cell clusters are color-coded according to the directional differences between the control ($n = 30$ patients) and MP ($n = 28$ patients) group. Directional differences were computed using a two-sided Wilcoxon rank-sum test (sign(r) -log(p-value), blue/red indicating an increased/ decreased frequency in the MP group; arrows point at cell clusters that differ most significantly between the patient groups). In cell clusters of the adaptive immune compartment (contoured in green), MP treatment resulted in decreased frequencies of CD4$^+$T$_{mem}$ at 1 and 6 h and CD4$^+$T$_{naïve}$ cells at 6 h, but no significant changes in CD8$^+$T or B cell frequencies. In cell clusters of the innate immune compartment (contoured in orange), MP treatment resulted in decreased frequencies of cMCs and M-MDSCs at 1 h, and ncMCs and mDCs at 1 and 6 h. In contrast, MP resulted in increased frequencies of CD56$^{lo}$CD16$^+$NK cells at 1 h and mDCs at 24 h. **b**, **c** Box-plots depicting the frequency of manually gated immune cell subsets corroborating observations contained in the immune cell atlas. Immune cell subsets for which MP's effect on adaptive (**b**) and innate (**c**) immune cell frequencies were most pronounced (CD4$^+$ T$_{mem}$, CD4$^+$ T$_{naive}$, cMCs) are shown. Neutrophil frequencies (not included in the immune cell atlas) are also shown. Box plots for all manually gated immune cells are available in Supplementary Fig. 4. All boxplots show median values, interquartile range, whiskers of 1.5 times interquartile range. (Two-sided Wilcoxon rank-sum test, *$p < 0.01$, **$p < 0.001$, ***$p < 0.0001$). Exact p-values are available in Supplementary Table 2.

cell cluster and signaling protein at each time point. The values derived in pre-surgical samples were subtracted from the values derived in post-surgical samples to infer the change of the net signaling responses to surgery. Inspection of the 2-dimensional cell atlas suggested that MP altered elements of the JAK/STAT and NF-κB pathways (Supplementary Fig. 3). With respect to the JAK/STAT pathway, the pSTAT3 and pSTAT5 (and to a lesser extent pSTAT1) responses were altered most prominently (Fig. 4, Supplementary Figs. 3, 5). In contrast, only small or no differences were observed for elements of the P38 and ERK/MAPK pathways including pP38, pERK1/2, pMAPKAPK2, prpS6, and pCREB responses (Supplementary Fig. 4).

The effects of MP on intracellular immune cell dynamics were signaling- and cell type-specific. With respect to the JAK/STAT pathway, MP resulted in a sustained attenuation of the STAT3 signaling response in adaptive immune cells, including CD4$^+$ T cell subsets (Tbet$^+$CD4$^+$T cells at 1 h and CD4$^+$T$_{mem}$ and CD4$^+$T$_{naïve}$ at 6 and 24 h after surgery) (Fig. 4a, b). Similarly, MP resulted in a prominent and early attenuation of the STAT5 signaling response in adaptive immune cells including

CD4$^+$ T cell subsets (CD4$^+$T$_{mem}$, CD4$^+$T$_{naïve}$, Tbet$^+$CD4$^+$T, and T$_{regs}$) and CD8$^+$ T cell subsets (CD8$^+$T$_{naïve}$ and CD8$^+$T$_{mem}$) at 1 and 6 h after surgery (Supplementary Fig. 5a, b). However, MP had little or no effect on the STAT3 and STAT5 signaling responses in innate immune cells, including neutrophils and MC subsets (Fig. 4a, c, Supplementary Fig. 5a, c).

In contrast, MP treatment resulted in elevated total IκBα (a canonical inhibitor of NF-κB) in both the innate and adaptive immune cell compartment. Differences were most pronounced 24 h after surgery for adaptive immune cell subsets (CD4$^+$T$_{mem}$, CD4$^+$T$_{naïve}$, Th1, T$_{regs}$, CD8$^+$T$_{naïve}$ and CD8$^+$T$_{mem}$), and some innate immune cell subsets (mDCs, cMCs, ncMCs, intMCs, and M-MDSCs). Differences for other innate immune cell subsets (neutrophils and NK cells) were already prominent 1 and 6 h after surgery (Fig. 5).

In summary, MP administration attenuate multiple intracellular signaling programs that are activated in response to surgery, including elements of the JAK/STAT and NF-κB signaling pathways. However, other important signaling pathways remain unaffected by MP. These included elements of the MAPK

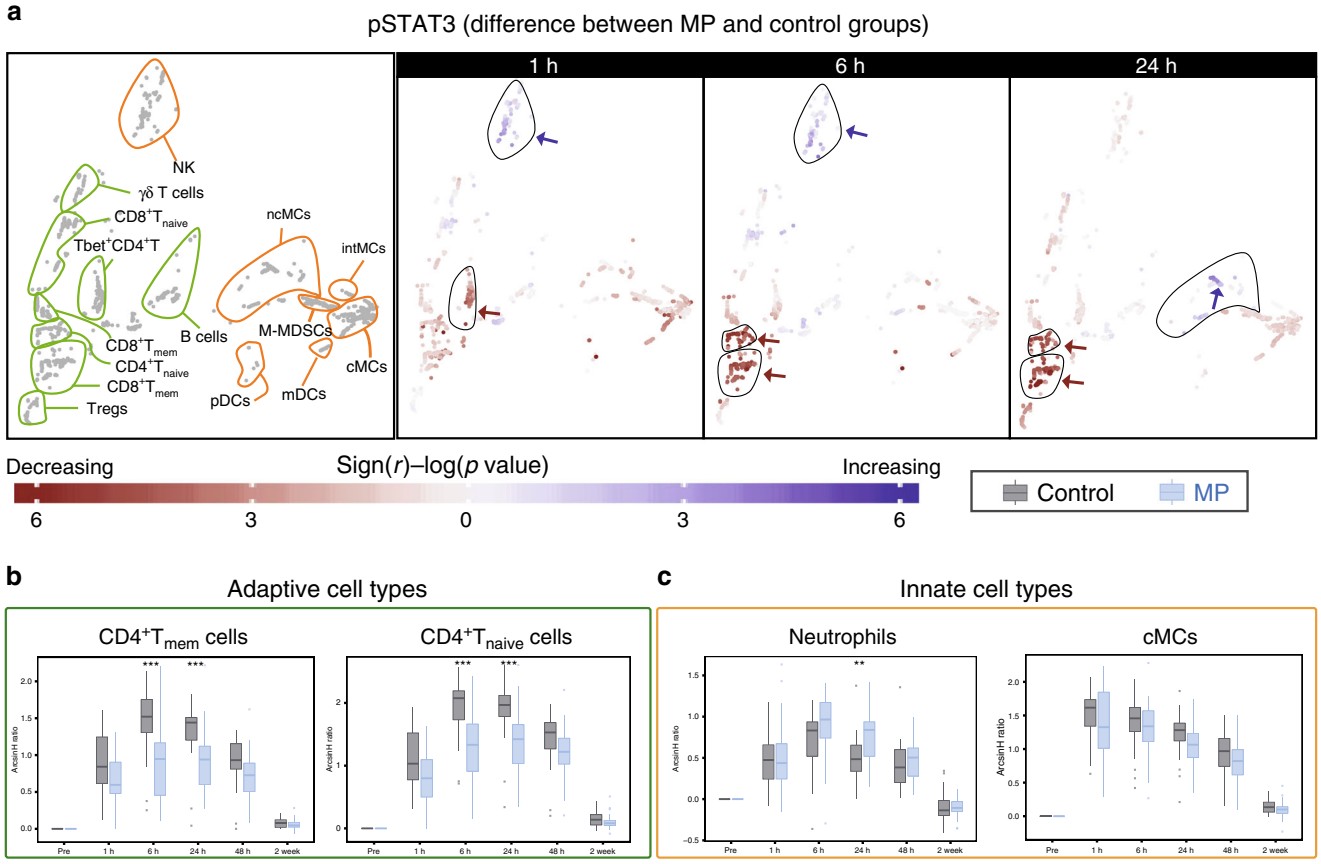

**Fig. 4 Alteration of intracellular pSTAT3 responses by MP. a** Immune cell atlas depicting differences of the phospho-(p)STAT3 response (arcsinh ratio) between the control ($n = 30$ patients) and MP ($n = 28$ patients) group at 1, 6, and 24 h after surgery relative to the preoperative time point. Blue/red cluster colors indicate increased/decreased signaling in the MP group, respectively (two-sided Wilcoxon rank-sum test). Arrows point at cell clusters that differ most significantly between the patient groups. In cell clusters of the adaptive compartment (contoured in green), MP treatment resulted in a sustained attenuation of pSTAT3 responses in CD4$^+$T cells (first in Tbet$^+$CD4$^+$T cells at 1 h, then in CD4$^+$ T$_{naive}$ and CD4$^+$ T$_{mem}$ cells at 6 and 24 h. In contrast, in clusters of the innate compartment (contoured in orange) MP treatment resulted in no significant changes in the pSTAT3 signal in monocyte subsets (including cMCs, intMCs, and M-MDSCs) and in only a modest increase in the pSTAT3 signal in CD56$^{lo}$CD16$^+$NK cells at 1 and 6 h and ncMCs at 24 h. **b, c** Box plots depict the pSTAT3 signal in manually gated immune cell subsets corroborating observations contained in the immune atlas. **b** MP's attenuation of the pSTAT3 signal was most pronounced in CD4$^+$ T cell subsets. **c** MP does not attenuate the pSTAT3 signal in neutrophils (signal is increased at 24 h) or cMCs. All boxplots show median values, interquartile range, whiskers of 1.5 times interquartile range (two-sided Wilcoxon rank-sum test, *$p < 0.01$, **$p < 0.001$, ***$p < 0.0001$). Exact $p$-values are available in Supplementary Table 2.

pathways such as pP38, pERK1/2, prpS6, and pCREB in all immune cell subsets as well as the STAT3 and STAT5 signaling pathways in innate immune cells, including neutrophils and MC subsets. Notably, signaling responses previously shown to strongly correlate with clinical recovery from pain and functional impairment after THA surgery[23,24], including STAT3 and CREB signaling responses in MCs, are unaffected by MP treatment. These findings raise the question as to whether MP administration affected these clinically important recovery parameters.

**MP does not alter pain, fatigue or functional impairment**. The course of postoperative pain, functional recovery of the operated joint, fatigue, and resulting impairment of daily functioning were captured over 4 weeks after surgery as previously described[23]. Instruments for assessing these clinical recovery outcomes included an adapted version of the Western Ontario and McMaster Universities Arthritis Index (WOMAC) and the Surgical Recovery Scale (SRS). No differences were detected between the MP and control groups (Fig. 6). These results are congruent with previous reports[23,24] and suggest signaling responses spared by MP treatment (such as STAT3 and CREB signaling in MCs) are more relevant determinants of the clinical recovery

parameters examined in this study than signaling responses that are affected by MP.

**Discussion**
This study provides an in-depth and functional profile of the effect of GCs on immune system dynamics after a major surgical trauma (THA). Analysis of patients' immune trajectories after surgery reveals that a single preoperative dose of 125 mg MP produces profound and cell-specific alterations of the innate and adaptive immune response for at least 48 h after surgery. Notably, MP treatment accentuates IκB signaling responses in all major immune cell subsets of the innate and adaptive compartments, while selectively inhibiting JAK/STAT signaling responses in the adaptive compartment only.

Interestingly, the modulation of the surgical immune response by MP does not affect assessed clinical recovery trajectories including pain, fatigue, and functional impairment of the operated hip. These results are in line with previous reports indicating that single-dose administration of GC in the perioperative period does not affect pain and functional trajectories beyond postoperative day 2, although some reports suggest short-term beneficial effects during the first 24–48 h[1,40–43]. The results are also

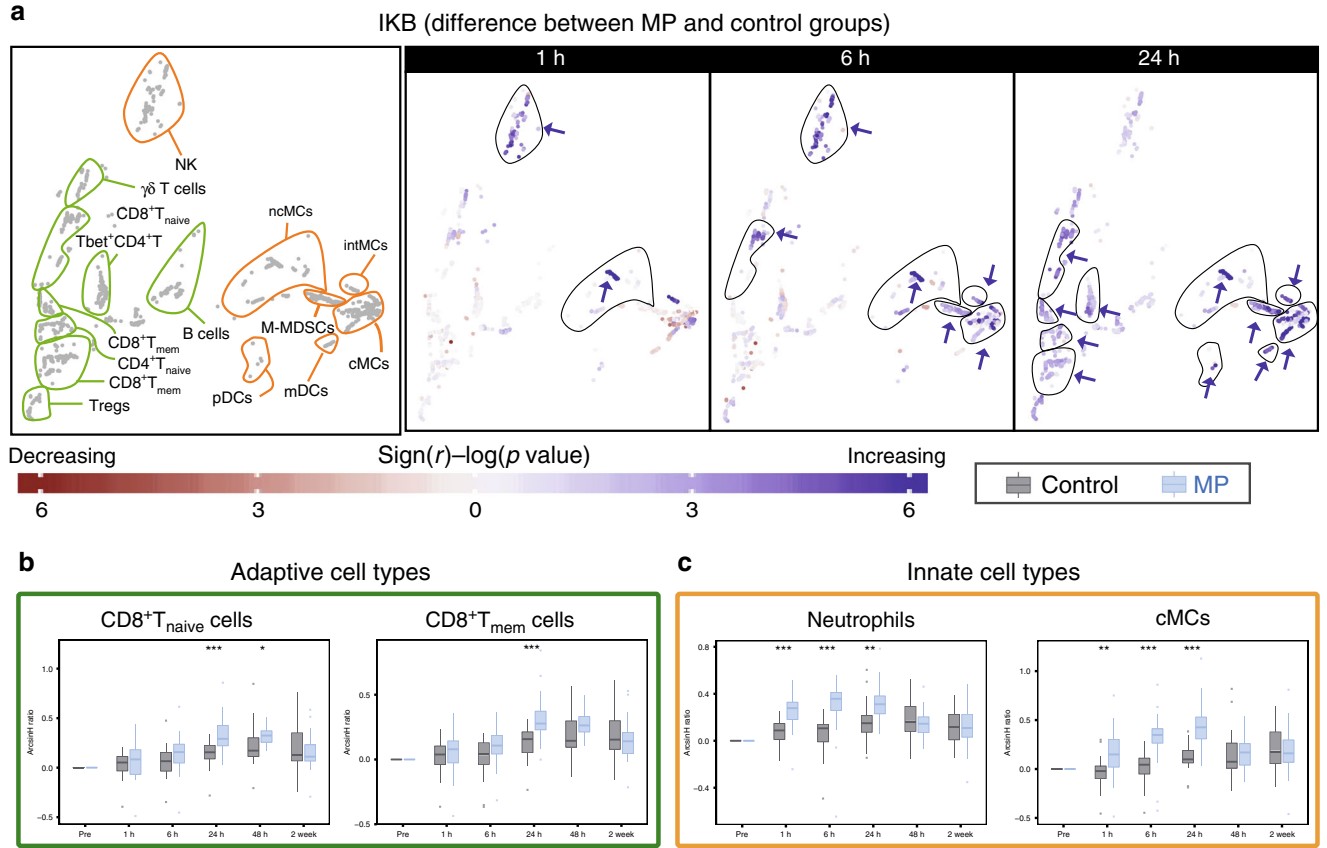

**Fig. 5 Alteration of total IκBα by MP. a** Immune cell atlas depicting differences in total IκBα (arcsinh ratio) between the control ($n = 30$ patients) and MP ($n = 28$ patients) group at 1, 6, and 24 h after surgery. Blue/red cluster colors indicate increased/decreased signaling in the MP group, respectively (two-sided Wilcoxon rank-sum test). MP treatment resulted in increased total IκBα in both the adaptive (CD4+T$_{mem}$, CD4+T$_{naive}$, CD8+T$_{naive}$ and CD8+T$_{mem}$, contoured in green) and the innate compartment (CD56$^{lo}$CD16+ NK cells, mDCs, cMCs, ncMCs, M-MDSCs and DCs, contoured in orange, which was most prominent at 6 and 24 h after surgery. **b, c** Box-plots depict total IκBα in manually gated immune cell subsets corroborating observations contained in the immune atlas. Select immune cell subsets for which MP's effect on total IκBα was most pronounced (CD8+ T$_{naive}$ and CD8+ T$_{mem}$, cMCs, and neutrophils) are shown. All boxplots show median values, interquartile range, and whiskers of 1.5 times interquartile range. (Two-sided Wilcoxon rank-sum test, *$p < 0.01$, **$p < 0.001$, ***$p < 0.0001$). Exact $p$-values are available in Supplementary Table 2.

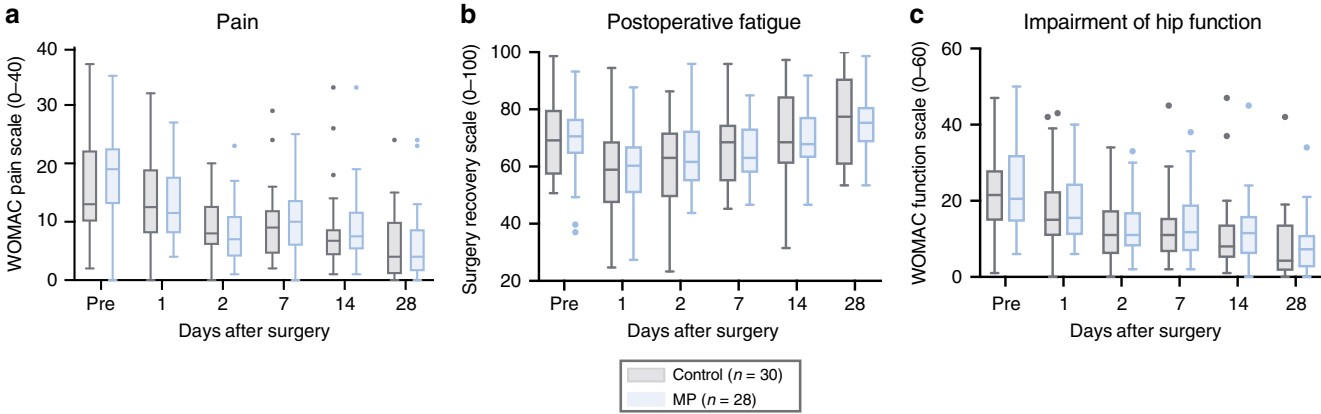

**Fig. 6 Clinical recovery outcome measures between control and MP treatment.** All boxplots show median values, interquartile range, whiskers of 1.5 times interquartile range of clinical recovery parameters (**a**) pain, (**b**) fatigue and daily functioning, and (**c**) functional impairment of the operated hip over the course of 28 days after surgery between control ($n = 30$ patients) and MP ($n = 28$ patients) groups (two-sided Wilcoxon rank sum test). Pain and functional impairment of the hip were assessed with an adapted version of the Western Ontario and McMaster Universities Arthritis Index (pain 0 to 40 = none to worst; function 0 to 60 = no to most severe impairment). Postoperative fatigue and daily functioning were assessed with the Surgical Recovery Scale (17 to 100 = worst to none).

consistent with the finding that signaling responses previously reported to strongly correlate with the resolution of pain and functional recovery (e.g. STAT3 and CREB in MCs) were not affected by MP[23]. In contrast, MP markedly inhibits STAT3 and STAT5 responses in CD4[+] and CD8[+] T cell subsets indicating that the perioperative use of GCs may alter other surgical recovery processes that specifically depend on JAK/STAT signaling in T cell subsets.

STAT3 and STAT5 are transcription factors that control multiple aspects of CD4[+] T cell differentiation (including Th1, Th17, and $T_{reg}$ differentiation) and CD8[+] T cell effector function implicated in adaptive immune responses to invading pathogens[44,45] and tumor surveillance[46,47]. After surgery, these host defense mechanisms must be balanced with effective wound healing mechanisms that require immunosuppressive cell activity exerted by regulatory T cells and myeloid-derived suppressor cells[48,49]. For example, Krall et al.[50] demonstrated in a recent rodent study, elegantly separating immune mechanisms activated by surgical trauma from mechanisms relevant for cancer surveillance, that immunosuppressive myeloid cells involved in wound repair also facilitated tumor growth, which is opposed by tumor-specific T cells[50]. Our finding that MP preferentially inhibits JAK/STAT signaling in T cells but not in myeloid cells after surgery raises the possibility that the perioperative use of GCs may negatively affect the balance between immune mechanisms required for wound repair and tumor surveillance, including tumor-specific T-cells. In our study, there was no report of postoperative infection or adverse wound healing in either patient group, consistent with the generally low incidence (<1%) of such adverse events in hip arthroplasty surgery[51]. As such, studies further examining the effects of perioperative GC administration on surgical outcomes that include wound healing, postoperative infections, and disease recurrence after cancer surgery seem warranted[2,13].

Mapping statistical inference information onto individual clusters of phenotypically defined immune cells complements the single-cell level characterization of MP's effect on the surgical immune response. Aspects of our findings are in agreement with prior immune profiling of GC administration in surgical and non-surgical patients[52,53]. For example, MP administration results in increased neutrophil and NK cell frequencies 6 and 24 h after surgery. This is consistent with previous reports documenting demargination and impaired extravasation of neutrophils and NK cells after GC treatment, which are partially mediated by the inhibition of L-selectin expression[52,54,55]. Conversely, MP administration resulted in decreased CD4[+] T cell frequencies (in particular CD4[+]$_{mem}$ T cells) 1 and 6 h after surgery, which is consistent with previous observations in surgical patients receiving GCs and the high sensitivity of T cells to apoptosis induction by GCs[52,56,57].

Similarly, some MP-mediated alterations of immune signaling responses echo previous findings. For example, MP increases total IκBα in the majority of innate and adaptive immune cells. These findings are consistent with in vitro[58] and in vivo studies, including a recent transcriptomic analysis of GC administration in healthy volunteers documenting direct induction of IκBα gene and protein expression via DNA-binding of the GC receptor[14,59].

Interestingly, MP has little effect on pNF-κB signaling (phospho-S529 on the p65 (RelA) subunit of NF-κB), which increases in innate immune cells after surgery, particularly in cMCs (Supplementary Figs. 3, 4). Our findings suggest that in the context of surgery, the MP-mediated increase in IκBα may primarily inhibit NF-κB signaling via cytoplasmic sequestration, rather than inhibiting phosphorylation at S529, a key phosphorylation events controlling NF-κB nuclear translocation in response to inflammatory signals such as IL-1β and TNF[60].

In addition, MP may inhibit NF-κB signaling through IκB-independent mechanisms, such as direct protein-protein interaction at NF-κB DNA binding sites[61,62]. However, such mechanisms would not have been detected with our current mass cytometry assay. These results emphasize the complexity and redundancy of NF-κB activation after traumatic injury. They also highlight the benefits of analytical platforms such as mass cytometry that allow probing multiple elements of the same pathway to gain a comprehensive understanding of the effect that GCs exert on immune cell function after surgical trauma.

Remarkably, MP treatment inhibits JAK/STAT signaling responses to surgery in adaptive immune cells, while minimally affecting these signaling responses (including inhibition of the pSTAT3, pSTAT5, and to a lesser extent, pSTAT1 signals) in innate immune cells. Several mechanisms likely underlie these findings, as the interaction between the GC receptor and the JAK/STAT signaling pathways is complex[63]. In general, GCs suppress the transcription of pro-inflammatory cytokines such as IL-2, IL-6, Interferons, and GM-CSF, which activate JAK/STAT signaling after injury. Inhibition of JAK/STAT cytokine production after trauma may differentially affect innate and adaptive cells as cytokine receptor expression and downstream activation of STATs is often lineage-dependent[45,64]. In vitro studies have also shown that GCs can stimulate the transcription of anti-inflammatory JAK/STAT-targeting cytokines such as IL-10, specifically in MCs but not in T cells[65]. As such, increased autocrine activation of JAK/STAT signaling by cytokines selectively induced by GCs in innate immune cells may account for observed preservation of JAK/STAT signaling responses in innate immune cells.

This study has certain limitations. The modest sample size recruited from a single clinic site and the restriction to THA surgery limit the generalizability beyond the studied population and type of surgery. In particular, given that our patient population is enriched for elderly patients (over 65-years old) we cannot conclude that age-related immune dysfunction such as immunosenescence did not contribute to the pronounced inhibitory effect of MP on adaptive cell signaling responses[66,67]. Similarly, only one single-dose GC regimen is examined. It is possible that repeated dosing of steroids may alter measured clinical outcomes. While this is presently not known, additional safety and efficacy data are warranted[68]. Future studies with larger and younger patient cohorts undergoing a broader array of surgical procedures and various GC administration regimens will be required to test the boundaries of the generalizability of our findings. While applied mass cytometry assays allow measuring over fifty parameters per cell, the number of assessed phenotypic markers and signaling responses is not exhaustive. In particular, our analysis does not include assessment of cytokine expression, cell migration, and apoptosis or proliferation, which would have complemented the functional assessment of proximal immune cell signaling responses. However, our approach, combined with emerging statistical methods allowing for integrated and multi-omic analysis of inflammatory responses, provides an analytical framework to expand upon in future studies simultaneously examining how GCs affect a patient's immunome, proteome, and transcriptome in the context of surgery.

GCs are commonly administered to patients undergoing surgery, but our understanding of their effect on patients' immune response is quite limited. We applied high parameter, single-cell mass cytometry at the bedside to produce a reasonably comprehensive atlas detailing the effects of MP on peripheral immune cell dynamics in patients undergoing major joint replacement surgery. MP treatment profoundly alters immunological trajectories after surgery, which is particularly pronounced for the adaptive immune compartment. However, cell-specific signaling

responses previously associated with critical clinical recovery parameters, including the resolution of pain and functional impairment, are spared by the MP treatment. These findings are consistent with the observation that MP does not alter these clinical recovery parameters. However, the pronounced effects of MP on the adaptive immune compartment call for studies of clinical outcomes potentially affected by such immune alterations. A particular intriguing field includes surgical oncology as GCs may affect immune mechanisms relevant for tumor surveillance.

## Methods

**Study design.** This double-blind, placebo-controlled, randomized, and prospective clinical study was approved by the Danish Health and Medicine Authority (EudraCT 2015-000102-19), the Ethics Committee for the Capital Region of Denmark (H-15007653, protocol approval July 2015), the Danish Data Protection Agency, and registered at ClinicalTrials.gov (NCT02542592). Patients were enrolled after obtaining written informed consent. Inclusion criteria were: age 55–80 years and the ability to speak and understand Danish. Exclusion criteria were: general anesthesia, cancer, autoimmune diseases including rheumatoid arthritis, allergy or intolerance to MP, local or systemic infection, continued systemic treatment with steroids within 30 days before surgery, insulin-dependent diabetes, atrial fibrillation, neurological diseases including Parkinson's, daily use of hypnotics or sedatives, alcohol use >35 units per week, active treatment of ulcers within 3 months before surgery, pregnancy, and breast-feeding or recent onset of menopause (<1 year) in women.

The manuscript was prepared according to the Consolidated Standards of Reporting Trials (CONSORT) recommendations for reporting randomized, controlled, clinical trials. The CONSORT chart is provided in Supplementary Fig. 1.

**Randomization and blinding.** A random allocation sequence (1:1 allocation rate, no block randomization) was created, and numbered and sealed envelopes were prepared to determine which arm of the study each patient would fall into. On the day of surgery, the envelopes were opened by a nurse not involved in any other aspect of the study, and either a single dose of 125 mg of methylprednisolone (Solu-Medrol®; Pfizer, Ballerup, Denmark) (MP group) or a single dose of isotonic saline (control group) was prepared in a separate room. MP or saline placebo were prepared in masked syringes and administered by one of two blinded investigators immediately after completion of spinal anesthesia. The dose and timing of MP administration were chosen based on a prior study suggesting beneficial effects on pain and recovery[40].

**Surgical and anesthetic procedure and anesthesia.** All patients underwent THA surgery for treatment of osteoarthritis at the Copenhagen University Hospital in Copenhagen, Denmark. Surgical and anesthetic procedures were previously described in detail[36]. All surgeries were performed under lumbar spinal anesthesia with 12.5–17.5 mg isobaric bupivacaine (5 mg/ml, 0.5%). After surgery, patients followed a routine, well-defined, fast-track rehabilitation regime, that included fluid therapy, a standard multimodal pain treatment, mobilization on the day of surgery, and well-defined discharge criteria[69].

**Surgical recovery outcomes.** Assessments were made 1 h before and 1, 2, 7, 14, and 28 days after surgery as previously described in detail[23]. In brief, fatigue and resulting functional impairment were captured with the Surgical Recovery Scale (SRS; 17–100 = worst/best score), a well validated questionnaire specifically designed for the surgical setting[70]. Pain and functional impairment of the hip were assessed with the Western Ontario and McMaster Universities Arthritis Index (WOMAC) adapted to the surgical setting[71]. Pain scores range from 0–40 (no/ worst pain), and function scores range from 0–60 (no/most severe impairment).

**Whole blood sample processing for mass cytometry.** Whole blood samples were collected in sodium-heparinized tubes at 6 time points (1 hour before surgery and 1, 6, 24, 48 h, and 2 weeks after surgery). Within 30 minutes of phlebotomy, samples (1 mL) were processed and fixed in Smart Tubes (Smart Tube Inc., San Carlos, CA), and then immediately stored at −80 °C. All samples were shipped on dry ice as a single batch to Stanford University (Stanford, CA) for further processing and analysis.

After thawing and erythrocyte lysis, samples were barcoded and stained with surface and intracellular antibodies using standardized protocols[23,38]. In brief, whole blood samples were subjected to erythrocyte lysis using Thaw-Lyse Buffer (Smart Tube, Inc., San Carlos, CA) and isolated leukocytes from each sample were treated ("barcoded") with a unique combination of 3 palladium isotopes (Trace Sciences, International, Wilmington, DE) in 0.02% saponin (Millipore-Sigma, St. Louis, MO). After barcoding, cells were pooled and treated in aggregate with 1:100 Human Fc block (Biolegend, San Diego, CA), stained with a custom panel of commercially available antibodies covalently conjugated to a proprietary polymer loaded with lanthanide isotopes (Fluidigm, Inc., South San Francisco, CA), then treated with an iridium-based DNA intercalator (Fluidigm, Inc., South San Francisco, CA)

(Supplementary Table 1). In order to minimize experimental variability, samples corresponding to an entire time series were barcoded, stained, and run simultaneously on the mass cytometry instrument[72,73]. In order to maximize the sensitivity of the assay for detection of differences between the MP and control groups, sample time series from patients in the MP group were randomly paired with samples from patients in the control group, and paired sample time series were barcoded and run using the same barcode plate. Barcoded samples were analyzed at a flow rate of ~600–800 cells/s. The output FCS files were normalized (https://github.com/nolanlab/bead-normalization/releases) and de-barcoded (https://github.com/nolanlab/single-cell-debarcoder/releases/tag/v0.2) using MatLab-based software[73,74]. The resulting FCS files were uploaded to the Cell Engine (https://cellengine.com, Primity Bio, Fremont, CA) flow cytometry analysis platform.

**Derivation of immune features.** Manual gating was performed using Cell Engine according to a standard gating strategy[38,75] (Supplementary Fig. 2). The following 21 cell types were included in the analysis: neutrophils, CD27$^+$ B$_{mem}$ cells, CD27$^-$B$_{naive}$ cells, CD56$^{hi}$CD16$^-$ NK cells, CD56$^{lo}$CD16$^+$ NK cells, CD4$^+$CD45RA$^-$T cells (CD4$^+$ T$_{mem}$), CD4$^+$CD45RA$^+$ T cells (CD4$^+$ T$_{naive}$), CD4$^+$Tbet$^+$ CD45RA$^-$T cells (Th1), CD4$^+$Tbet$^+$CD45RA$^+$ T cells, CD25$^+$FoxP3$^+$CD4$^+$ T cells (T$_{regs}$), CD8$^+$CD45RA$^-$T cells (CD8$^+$ T$_{mem}$), CD8$^+$CD45RA$^+$ T cells (CD8$^+$ T$_{naive}$), CD8$^+$Tbet$^+$CD45RA$^-$ T cells, CD8$^+$Tbet$^+$CD45RA$^+$ T cells, TCRγδ T cells, CD14$^+$CD16$^-$ classical monocytes (cMCs), CD14$^-$CD16$^+$ non-classical monocytes (ncMCs), CD14$^+$CD16$^+$ intermediate monocytes (intMCs), monocytic myeloid-derived suppressor cells (M-MDSCs), myeloid dendritic cells (mDCs), and plasmacytoid dendritic cells (pDCs).

Cell frequencies for mononuclear cells were expressed as a percentage of gated singlet live mononuclear cells (cPARP$^-$CD45$^+$CD66$^-$), while frequencies for neutrophils were expressed as a percentage of gated singlet live cells (cPARP$^-$). Frequencies were calculated at the pre-surgical time point, and at 1, 6, 24, 48 h, and 2 weeks after surgery.

The intracellular expression of the following functional markers was simultaneously quantified per single cell: phospho-(p)STAT1, pSTAT3, pSTAT5, pSTAT6, pNF-κB, pMAPKAPK2, pP38, prpS6, pERK1/2, pCREB, and total IκBα. For each cell type, basal signaling activities were expressed as the median signal intensity (arcsinh transformed value) of each signaling protein. Signaling changes in response to surgery were expressed as the difference in median signal intensity (arcsinh ratio) from baseline signaling.

All p-values calculated from manually gated cells were derived using a two-sided Wilcoxon rank sum test and are contained within Supplementary Table 2.

**Visualization of immune system dynamics.** To visualize trajectories of the immune system over time, all features extracted using manual gating were divided into "adaptive" and "innate" groups based on prior knowledge. All features were normalized by subtracting the value of the pre-surgical time point. All data points were visualized using two separate dimension-reduction analyses using the Isomap algorithm[76]. To produce continuous projections between the data points, a linear transformation was used after dimension reduction.

**Bootstrapped clustering methods.** To complement the manual gating analysis, an unsupervised clustering approach was used to partition cells into phenotypically distinct subpopulations. State-of-the-art algorithms, such as Citrus[77], FlowSOM[78], and PhenoGraph[79] produce variable results across algorithm runs which lead to inconsistent cluster-based features that can result in unstable classification results. To address these limitations, we implemented a bootstrapped clustering and classification pipeline enabling us to identify the key cell types and signaling pathways that differ between the control and MP groups. Notably, the clustering approach is downsampling-free and can efficiently integrate all cell events over a large number of samples. This is achieved through a metaclustering strategy where all cells from each sample are first clustered independently to define sample-specific clusters. The cluster centers are then integrated to define a set of metaclusters across all samples. This process is computationally efficient and can scale to a large number of samples.

To account for the variability that arises across individual runs of a clustering algorithm, CD45$^+$ cells across all 331 samples were subjected to bootstrapped meta-clustering (CD66$^+$/CD45$^-$ neutrophils were excluded of the analysis for ease of representation, and examined separately). Each of the refined FCS files corresponding to each sample were then coarsely clustered using k-means based on the 26 phenotypic markers. The number of cluster centers input to k-means for each sample FCS files was $\sqrt{N/2}$, where $N$ is the number of cells in the FCS file. The resulting cluster centers associated with each FCS file were then extracted and concatenated to form a new data matrix. This matrix of cluster centers was then repeatedly clustered using k-means into 30 meta-clusters 200 different times to account for the variability between clustering iterations. Features encoding cell frequencies and functional marker expressions were constructed for each individual cluster across all meta-clustering iterations.

To compute the frequency of a cell cluster in a particular sample, we calculate the proportion of that sample's cells assigned to that cluster as a percent of the total cells in that sample. The frequencies used in downstream bioinformatics analyses are thus normalized by the total number of events in each sample's FCS file. To

compute signaling responses of individual cell clusters at each post-surgical time point for each cell cluster, the difference in signal intensity of each signaling maker between the postoperative time point and the preoperative time point (Arcsinh ratio) is quantified.

**Random Forest analysis**. A RF classifier[80] was trained at each time point using the cell frequency and signaling-based features constructed for each of the identified meta-clusters. At each individual time point, these features were used in a repeated leave-group-out cross validation approach to predict the probability that each sample came from a patient in the MP group. The leave-group-out cross validation pipeline is an ensemble-based classification approach, where a model training and prediction procedure is repeated over 200 iterations. At each iteration, half of the samples were used to train a RF model and predictions were made on samples in the remaining half of the data. The ultimate predicted value for a sample was the median predicted probability over the predictions from the cross-validation iterations where the sample was included in the test set. The predicted probabilities for each sample were used to construct ROC curves with the associated area under the ROC curve (AUC) using the pROC package in R. The p-value (p) from a Wilcoxon Rank Sum Test was used at each time point to test the null hypothesis that the predicted probabilities for the MP and control were equal.

**Visualization of immune cell atlas**. The bootstrapped meta-clusters were visualized in two dimensions using PCA. Each cluster was first represented by the median expression of its 26 phenotypic markers and then reduced to a two-dimensional representation through PCA. Clusters colored by their median phenotypic marker expression were used to annotate cell populations. The two-dimensional PCA plot of all identified clusters provided the backbone of a high-resolution immune cell atlas that can be used to communicate statistical information about cell frequency and signaling differences between control and MP patients.

Best p-value plots provide a visual depiction of the relative effect of MP on individual immune cell clusters at each of the time points. All of the frequency and signaling-based features computed for each of the identified clusters were used to statistically test for differences between control and MP patients. In every cluster and for every frequency and signaling-based feature, a Wilcoxon rank sum two-sided test was used to test the null hypothesis that the mean value of the particular feature was equal between control and MP samples in the cluster. The corresponding $-\log_{10}$ p-value was recorded and points (corresponding to clusters) were ultimately colored by the best $-\log_{10}$ p-value across all tested features. A plot with clusters colored by their best p-value was created for the 1, 6, 24, 48 h, and 2 week time points.

Sign(r) $-\log_{10}$ p-values plots were constructed to communicate whether the mean value of each feature (frequency or signaling feature) was higher among control or MP samples within each cluster. For a given cluster-based feature, f, we computed the mean value of f in the MP and control samples. We defined sign(r) to be 1 if the mean value of the feature was higher in the MP group or −1 if the feature was higher in the control group. We used a Wilcoxon Rank Sum test to test the null hypothesis that the mean of f was the same between MP and control groups in the particular cluster. The Wilcoxon Rank sum test yielded a corresponding p-value for each cluster and each cluster was colored according to the sign(r)-$\log_{10}$ p-value. Blue/red colors indicate features that are higher/lower in the MP compared to the control group.

**Reporting summary**. Further information on research design is available in the Nature Research Reporting Summary linked to this article.

## Data availability
Raw data were uploaded and made publicly available at https://flowrepository.org/id/FR-FCM-Z2AT. The source data underlying Figs. 3b, c, 4b, c, 5b, and c are provided as a source data file. Source data are provided with this paper.

## Code availability
The predictive modeling was performed using the VoPo framework[81], available at: https://nalab.stanford.edu/vopo/. Scripts and processed data for reproduction of the results are available at https://github.com/stanleyn/steroid_immune_data. Source data are provided with this paper.

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

## Acknowledgements

This work was supported by National Institutes of Health Grant 1K23GM111657 (to E.G. and B.G.), the Department of Anesthesiology, Perioperative and Pain Medicine at Stanford University (to E.G., M.S.A., N.A., and B.G.), Stanford Immunology Training Grant (5T32AI07290-33) Stanford Anesthesia Training Grant (T32GM089626) (to N.S.), Fondations des Gueules Cassees (to F.V.), NIH T32 GM 089626 (to K.K.R.), German Research Foundation (STE2757/1-1) (to I.A.S.). The work was also partially supported by

NIH R01AG058417, R01HL13984401, R01DA050960, R61NS114926, R21DE02772801, and KL2TR003143).

## Author contributions

E.G. contributed to sample processing and mass cytometry analysis, data processing and interpretation, and manuscript preparation and editing. N.S. contributed to meta-clustering analysis, data processing and interpretation, and manuscript preparation and editing. V.L-.L. contributed to study design, patient recruitment and assessment, sample collection, and manuscript preparation and editing. J.E., A.T., and F.V. contributed to data processing and interpretation and manuscript preparation and editing. A.C., S.G., and R.F. contributed to analysis of the mass cytometry dataset. K.R., I.S., and D.G. contributed to data interpretation and manuscript preparation and editing. E.T. and B.C. contributed to sample processing and data analysis. H.K. contributed to study design, patient recruitment and assessment, and manuscript preparation and editing. M.A. contributed to study design and manuscript preparation and editing. N.A. contributed to mass cytometry data analysis and manuscript preparation and editing. B.G. contributed to all aspects of the study including study design, data processing and analysis, and manuscript preparation and editing.

## Competing interests

The authors declare no competing interests.
