## [Peer Review File · Nature Communications]

Reviewers' comments:

Reviewer #1 (Remarks to the Author):

In this study, the group used a cohort of surgical patients to examine the dynamics of how surgical trauma alters the circulating immune cell subset landscape with a comparison of the effect of methylprednisolone (MP) treatment on immune cell phenotypes. They used 11 phospho-signaling mediators as functional cellular readouts for how MP changes innate and adaptive immune cell subsets. The study uses an appropriate and well thought out data analysis workflow to reduce the complex high-dimensional single cell analysis by mass cytometry into interpretable results. In particular, the random-forest algorithm worked well to demonstrate that the majority of significant changes in cellular phenotypes occurred within 48 hours after surgical trauma. There are several new contributions to the trauma immunology field that have basic science and clinical implications. First, MP had the most significant effect on cytokine signaling responses by adaptive immune cell types and did not markedly change innate cells (except for I κ B α): Second, most of the highly significant circulatory immune changes to trauma occurs within 3 days after the insult. This is consistent with already published work for surgical trauma and in trauma patients.

The mass cytometry data analysis is a strength of the study. Another strength is that the MP treatment had a significant effect on specific signaling pathways in adaptive immune cell types. The work will be of general interest to the surgery and trauma field and does contribute novel information. There are some issues that should be addressed for clarity and to increase the impact of the study:

1. The format of the cell clusters in 2-dimensional space is somewhat hard to see, especially with the signaling studies.
2. Along this same issue, how did they generate these cluster plots? Were they down sampled and how many events were used for the clusters. Also, are the frequencies relative or normalized to be similar as event dots? These 2-dimensional plots are not what is normally seen in mass cytometry or single-cell analysis studies, so clarifying how these were generated and associated information should be included in the manuscript.
3. The clinical finding was that MP treatment had no measurable effect on recovery or post-operative pain conditions. The discussion section suggests that giving MP treatment to patients after surgery may not be a good idea given that the treatment inhibits adaptive immune cells and that might have detrimental effects on host immune function – in particular, post-operative infections. Did the authors examine post-operative infection rates in the control and MP treated patients? This should be indicated in the manuscript text or added as a clinical outcome.
4. What positive controls do the authors use to validate their signaling antibody stains with the smart tube approach that they are using?
5. In Supplemental Figure 2, it is difficult to see the arrows, please increase their size or make the arrow ends bigger
6. The CD7 marker plot in the gating scheme is confusing – it indicates NK cells that are CD7+/CD14-. Is this an error in the figure? Usually CD56 or CD94 is used on CD3- cells to delineate NK cells? This review thinks that CD7 is on T cells, but could be wrong.
7. Supplemental Figure 4 is unreadable. The authors need to figure out a better way to present these data. This is challenging but needed, the figure cannot be enlarged with good resolution either.

James Lederer

Reviewer #2 (Remarks to the Author):

In this manuscript, the authors describe a mass cytometry approach to study changes in the immune cell landscape and relevant signal transduction pathways after a single dose of glucocorticoids (GCs)

applied to patients before surgery for total hip replacement. In particular, the authors analyzed the abundance of a variety of innate and adaptive immune cell subtypes in the blood and characterized the activity of the JAK/STAT, NFkB and MAPK pathways at several time points after surgery. The experiments are technically sound and the work provides a comprehensive overview of the changes induced by GCs in immune cells. However, the study has also its limitations and suffers from the fact that the observed alterations were not linked to differences in clinical outcome. There are a number of concerns related to this study which need to be addressed and/or discussed.

1. The study includes only patients of 55-80 years of age, which is of course intrinsic to the study design since very few young individuals will undergo a hip replacement. Nevertheless, this feature brings about that many of the test persons presumably have an impaired immune system due to immunosenescence. Therefore it is unclear to which extent the findings will reflect the situation of a fully functional immune system present in healthy young individuals. Rather, some of the observations may be related to the decreased activity and abundance of specific immune cell subsets in elder individuals.

2. Supplementary Figure 4 is much too small and should be subdivided into several separate figures. Currently, it is impossible to read and interpret the data contained therein.

3. It is noteworthy that several findings described in this manuscript are trivial and have been known for a long time. For instance, the observation that CD4+ T cells rapidly decrease after MP treatment simply reflect the high sensitivity to apoptosis induction of this cell type, a process that has been studied for decades. Similarly, the fact that the number of neutrophils rapidly increases after MP injection is also well known and is a consequence of demargination mediated by altered expression of adhesion molecules.

4. Inherent to the analytical method, the reported results focus on the abundance of immune cells in the blood and the activity of signal transduction pathways. No information, however, is provided on processes such as proliferation, apoptosis, migration and adhesion. Thus, the mechanistic insights are limited although the information is still important.

5. An interesting finding is that MP inhibits Stat3/5 phosphorylation in T cells but almost not in neutrophils and monocytes. In contrast, IκBα is upregulated in both innate and adaptive immune cells. However, it is possible that this and other observations are related to the immunosenescence in the analyzed cohort of elder patients rather than being a general phenomenon. It would thus be important to validate some of the data in a cohort of younger patients with a different diagnosis but also treated with MP to check whether similar changes apply to individuals with a fully functional immune system.

6. The authors should comment on whether the single dose of MP had any effect on post-surgery vomiting and nausea compared to the placebo group. Of note, this effect of GCs is not linked to their anti-inflammatory activity as suggested by the authors in the introduction.

7. Could it be that the lack of efficacy of MP on the clinical outcome, namely pain and functional recovery 4 weeks after surgery, is due to the fact that GCs were only injected once? Would the outcome be different if GCs were repeatedly administered instead? This issue should be discussed in more detail.

8. The induction of IκBα is only one out of several mechanisms by which GCs inhibit NFkB signaling. As a matter of fact, interaction of the GC receptor with NFkB in the nucleus via protein-protein-interaction is believed to be the more relevant mechanism in this context (e.g. reviewed in *Nat Rev Mol Cell Biol* 2017, 18:159-174). Such an effect, however, cannot be addressed by the approach used

in this manuscript. Interpretations concerning the impact of GCs on NFkB signaling should thus be made with caution.

Reviewers' comments:

Reviewer #1 (Remarks to the Author):

1- *The format of the cell clusters in 2-dimensional space is somewhat hard to see, especially with the signaling studies.*

- We appreciate the reviewer's comment. In the revised manuscript, to facilitate the visualization of statistically significant cell clusters, we have contoured the immune cell subsets that contain cell clusters that differ the most between the MP and control groups (revised **Fig. 3-5, Fig. S5**). The primary purpose of the generated plots is to be able to quickly identify cell populations and signaling activity that are significantly different between control and MP groups. By design, the bootstrapped clustering algorithm produces a collection of similar although independent clusters. As a result, these clusters tend to exhibit

some overlap when projected into two dimensions. Additional information and justification for the choice of the layout is provided in our answers to comment #2.

2- *Along this same issue, how did they generate these cluster plots? Were they down sampled and how many events were used for the clusters. Also, are the frequencies relative or normalized to be similar as event dots? These 2-dimensional plots are not what is normally seen in mass cytometry or single-cell analysis studies, so clarifying how these were generated and associated information should be included in the manuscript.*

- We thank the reviewer for identifying areas that need further clarification regarding the specifics of data processing. In the revised manuscript, we provide additional information related to the algorithm used to generate cluster plots (see methods section p. 5, para. 2, 4; source code p. 6, para. 5; and results section p. 7, para. 4).

The purpose of this algorithm was to compute statistics that quantify the differences between patient groups and to allow communicating group-level statistics on a per-cluster basis. Currently, there are two main approaches for visualizing and comparing mass cytometry data across multiple samples. Dimensionality reduction algorithms, such as tSNE and UMAP, are used to project a subset of cells across samples onto two dimensions. However, high dimensionality reduction algorithms that depict individual cells on a 2D plot do not allow visualization of cell type-specific statistics. Alternatively, clustering methods such as standard stochastic clustering algorithms (such as k-means, FlowSOM, or PhenoGraph) focus on automated cell population discovery. However, these algorithms require subsampling of a random subset of cell events, thereby producing different solutions with each sampling. As features are engineered from the generated clusters for predictive analyses (such as Random Forest analysis in Fig. 2C), prediction accuracy can vary significantly depending on the clustering solution used.

A major benefit of our clustering approach is that it does not require subsampling a random subset of cells. Instead, it uses repeated parallel analysis of all cell events to produce a set of stable clusters (or cell populations). The algorithm effectively counteracts the variability associated with re-sampling of random cell subsets and thereby facilitates deduction of more robust predictions (**Fig. 2**, above).

We also would like to emphasize that findings visualized using our clustering algorithm have been validated with a manual gating approach to ensure that these findings are not inherent to the clustering approach (Bar graphs in **Fig. 3-5, Fig. S4, Fig. S5**).

Specific questions raised by the reviewer are addressed below:

- *How did they generate these cluster plots?* To generate the cluster plots, the first step is to obtain multiple clustering results across all samples, resulting in a common set of ‘bootstrapped clusters’ across all samples. This is achieved by running our meta-clustering approach multiple times. A description of this repeated meta-clustering procedure is found in the Methods section entitled ‘Bootstrapped clustering methods’ (p. 5 para 2).

After generating multiple clustering solutions, we project each of the clusters into two dimensions using principal component analysis (PCA) based on the expression of all phenotypic markers (see Methods: Visualization of immune cell atlas). Note that we could have visualized the clusters as a minimum spanning tree in a manner similar to FlowSOM³ or SPADE⁴ but we found that PCA works best to preserve phenotypic similarities/differences between clusters. Additionally, due to the nature of the repeated clustering algorithm, the produced clusters are distinct yet highly correlated. Therefore, a minimum spanning tree visualization would tend to put the highly correlated clusters (e.g. phenotypically very similar) on the same branch and make visual interpretation difficult.

Next, we calculate statistics for each cluster that reflect the difference between groups (see Methods: Visualization of Immune Cell Atlas). Per-cluster statistics are calculated for both frequency and functional marker expression. Our statistics are based on a function of the Wilcoxon Rank Sum test p-value. Therefore, we can color each cluster in the two-dimensional plot based on these computed values.

- *Were they down sampled and how many events were used for the clusters?* An advantage of our meta-clustering approach is that we can efficiently cluster all cells across a large number of samples. Therefore, downsampling is not required. As a result, we were able to use all of the events from each individual FCS file. We have clarified this point in the Methods section (p.5, para.2).

- Are the frequencies relative or normalized to be similar as event dots? When computing frequencies for a particular cluster, we calculate the proportion of that sample's cells assigned to that cluster. Therefore, the frequencies used in downstream bioinformatics analyses are normalized by the total number of events in each sample's FCS file. We specify that cell frequencies are normalized in the revised method section (p.5, para.4)
- 3- The clinical finding was that MP treatment had no measurable effect on recovery or post-operative pain conditions. The discussion section suggests that giving MP treatment to patients after surgery may not be a good idea given that the treatment inhibits adaptive immune cells and that might have detrimental effects on host immune function – in particular, post-operative infections. Did the authors examine post-operative infection rates in the control and MP treated patients? This should be indicated in the manuscript text or added as a clinical outcome.
- We thank the reviewer for this suggestion. However, the incidence of postoperative infection after hip-replacement surgery is generally very low (< 1%).⁵ Not too surprisingly, no postoperative infections were reported in our patient cohort. We provide this information in p.10, para.1 of the revised manuscript.
- 4- What positive controls do the authors use to validate their signaling antibody stains with the smart tube approach that they are using?
- For positive controls in the validation of signaling antibodies, we use whole blood stimulated with LPS (expected positive signal for pERK1/2, pP38, pMK2, pCREB, pNF-κB and IκB degradation in TLR4-expressing innate immune cell subsets, such as classical monocytes, cMCs), or Interferon alpha (expected positive signal for pSTAT1, 3, 5, 6 in innate and adaptive cells, such as cMCs and CD4⁺ T cells). Samples were fixed and processed using the Smart Tube system as for patient samples. Antibodies are titrated to obtain a maximal arcsinh ratio (difference between the stimulated and unstimulated condition) of the phospho-specific signal measured in classical monocytes. An exemplary result highlighting positive and negative control experiments is depicted in **Fig. 3** of our answers to the reviewers' comments.

Figure 3. Positive controls used for intracellular signaling antibody signal titration and validation. Whole blood samples from a healthy volunteer was left unstimulated (*first row*) or stimulated with LPS alone (1 μg/mL, *second row*), IFN-α (100 ng/mL, *third row*) for 15 min and analyzed using mass cytometry. The heat map is colored according to the arcsinh ratio of the intracellular antibody signal in the stimulated vs. unstimulated condition). Expected positive signals are observed for pERK1/2, pP38, pMAPKAPK2, pS6, pCREB, pNF-κB and total IκB in response to LPS in innate immune cells expressing TLR4 (such as cMCs), but not in immune cells that do not express, or express low levels of TLR4 (i.e. CD4⁺ T cells). Similarly, expected positive signals are observed for pSTAT1, pSTAT3, pSTAT5, and pSTAT6 in response to IFN-α in innate and adaptive immune cells.

- 5- In Supplemental Figure 2, it is difficult to see the arrows, please increase their size or make the arrow ends bigger
- We have increased the arrows' size in the revised **supplemental Figure 2**, as requested.

6- *The CD7 marker plot in the gating scheme is confusing – it indicates NK cells that are CD7+/CD14-. Is this an error in the figure? Usually CD56 or CD94 is used on CD3- cells to delineate NK cells? This review thinks that CD7 is on T cells, but could be wrong.*

- The reviewer is correct that CD7 is expressed on T cells. However, CD7 is also highly expressed on NK cells and as such, is a useful lineage marker used prior to CD56/CD16 gating to distinguish NK cells (which are CD56^{+/lo}, CD16^{+/-} and CD7⁺) from contaminating CD56⁺ monocytes and dendritic cells as well as CD16⁺ monocytes (which are CD7 negative), as described in the manuscript by Milush et al. (Blood, 2009).⁶ In our gating strategy for NK cell subsets, CD7 is used subsequent to gating out CD3⁺ T cells. In revised Fig. S2, we also replaced NK cells (CD7⁺CD14⁻) by NK cells (CD7⁺) as CD14 is not utilized in the gating of NK cells.

7- *Supplemental Figure 4 is unreadable. The authors need to figure out a better way to present these data. This is challenging but needed, the figure cannot be enlarged with good resolution either.*

- We thank the reviewer for this suggestion. We have reorganized the representation of the data in Fig. S4 and grouped individual bar graphs by frequency or functional attribute, which are available in high resolution as in the revised supplemental material (revised **Fig. S4A-K**).

Reviewer #2 (Remarks to the Author):

1. Comment related to age of study population: *The study includes only patients of 55-80 years of age, which is of course intrinsic to the study design since very few young individuals will undergo a hip replacement. Nevertheless, this feature brings about that many of the test persons presumably have an impaired immune system due to immunosenescence. Therefore it is unclear to which extent the findings will reflect the situation of a fully functional immune system present in healthy young individuals. Rather, some of the observations may be related to the decreased activity and abundance of specific immune cell subsets in elder individuals.*

Additional comment regarding immunosenescence: An interesting finding is that MP inhibits Stat3/5 phosphorylation in T cells but almost not in neutrophils and monocytes. In contrast, IκBα is upregulated in both innate and adaptive immune cells. However, it is possible that this and other observations are related to the immunosenescence in the analyzed cohort of elder patients rather than being a general phenomenon. It would thus be important to validate some of the data in a cohort of younger patients with a different diagnosis but also treated with MP to check whether similar changes apply to individuals with a fully functional immune system.

- We agree that our patient population is enriched for older patients. However, the elderly population constitutes an important clinical age cohort, as a large and increasing fraction of elderly patients undergo surgery (close to 50%). This population is particularly susceptible to postoperative complications.⁷ We fully agree with the reviewer though that our findings may not generalize to a younger population because of immunosenescence related alterations of immune function.^{8,9} We address this in greater details when discussing study limitations (p. 11, para. 2).
- In our patient cohort, examining the potential consequences of immunosenescence within the age range of studied patients can be explored in limited ways when excluding patients older than 65 years. We specifically examined STAT3/STAT5 responses, which were decreased in CD4⁺/CD8⁺ T cells, but not in cMCs, in patients treated with MP. Our analysis revealed that reported cell-specific STAT3/STAT5 inhibition in CD4⁺ and CD8⁺ T cells, but not in cMCs, was preserved when excluding patients older than 65 years. The results suggest that immune dysfunction aggravated with increasing age is not a major confounder of the observed cell-specific effect of MP. These results are summarized in **Fig. 4** of our answers to the reviewers' comments.

Figure 4: pSTAT3/pSTAT5 signal in classical monocytes, CD4⁺ and CD8⁺ T cells in patients younger than 65 year-old. The results show that cell-specific inhibition of pSTAT3 in CD4⁺ T cells and pSTAT5 in CD8⁺ T cells observed for the entire patient population is preserved when restricting the dataset for patients < 65 years old (u65). *** p<0.001 Mann-Whitney test.

2. *Supplementary Figure 4 is much too small and should be subdivided into several separate figures. Currently, it is impossible to read and interpret the data contained therein.*

We thank the reviewer for this suggestion and have organized **Fig. S4** into separate panels according to cell attributes (frequency or signaling responses), with higher resolution figures (revised **Fig. S4A-K**).

3. *It is noteworthy that several findings described in this manuscript are trivial and have been known for a long time. For instance, the observation that CD4⁺ T cells rapidly decrease after MP treatment simply reflect the high sensitivity to apoptosis induction of this cell type, a process that has been studied for decades. Similarly, the fact that the number of neutrophils rapidly increases after MP injection is also well known and is a consequence of demargination mediated by altered expression of adhesion molecules.*

- We agree with the reviewer that some of our findings are confirmatory in nature, particularly observed changes in cell frequencies of CD4⁺ T cells and neutrophils. While they are not novel, they are important as they speak to the rigor of the study (“positive controls”). We refer to these previous studies and have emphasized this point in the discussion of the revised manuscript p. 10, para. 2.

4. *Inherent to the analytical method, the reported results focus on the abundance of immune cells in the blood and the activity of signal transduction pathways. No information, however, is provided on processes such as proliferation, apoptosis, migration and adhesion. Thus, the mechanistic insights are limited although the information is still important.*

- We agree that the current assay does not allow assessment of immune cell proliferation, apoptosis, migration, and adhesion. We highlight this limitation in the discussion section of the revised manuscript (p. 11, para. 2). Assays assessing proliferation, apoptosis, migration and adhesion typically require collection, purification and preservation of live PBMCs. Results of these assays can therefore be significantly confounded by sample processing. In contrast, our study focused on whole blood samples fixed immediately after sampling to keep the confounding influence of sample processing to a minimum. As such, one rationale for focusing on the analysis of whole blood was to capture immune cell behavior (intracellular signaling responses) as they occur *in vivo*. A second important rationale was consideration of previous results of such analyses that were strongly associated with surgical recovery outcomes.¹⁰

5. *The authors should comment on whether the single dose of MP had any effect on post-surgery vomiting and nausea compared to the placebo group. Of note, this effect of GCs is not linked to their anti-inflammatory activity as suggested by the authors in the introduction.*

- Previous studies, including a study from Dr. Kehlet's group (a co-author)¹¹ have thoroughly documented that a single dose of MP significantly decreases PONV.^{12,13} As such, we did not systematically collect data to capture occurrence and severity of postoperative nausea and emesis. We agree with the reviewer that GC's effects on PONV are likely not related to their anti-inflammatory activity. We have edited the respective sentence in the revised document for clarification (p. 2, para. 1).
6. *Could it be that the lack of efficacy of MP on the clinical outcome, namely pain and functional recovery 4 weeks after surgery, is due to the fact that GCs were only injected once? Would the outcome be different if GCs were repeatedly administered instead? This issue should be discussed in more detail.*
- The reviewer raises an important point. It is possible that repeated dosing of steroids may alter measured clinical outcomes¹⁴ although more efficacy and safety data are required. We discuss this point in more details in the limitation section of the revised manuscript (p. 11, para. 2).
7. *The induction of IκBa is only one out of several mechanisms by which GCs inhibit NFκB signaling. As a matter of fact, interaction of the GC receptor with NFκB in the nucleus via protein-protein-interaction is believed to be the more relevant mechanism in this context (e.g. reviewed in Nat Rev Mol Cell Biol 2017, 18:159-174). Such an effect, however, cannot be addressed by the approach used in this manuscript. Interpretations concerning the impact of GCs on NFκB signaling should thus be made with caution.*
- We agree with the reviewer and discuss this alternative mechanism of GC-mediated NFκB signaling modulation in more details in the revised manuscript (p. 10, para. 4).^{15,16}

References

- 1 Bendall, S. C., Nolan, G. P., Roederer, M. & Chattopadhyay, P. K. A deep profiler's guide to cytometry. *Trends in immunology* **33**, 323-332, doi:10.1016/j.it.2012.02.010 (2012).
- 2 Bendall, S. C. *et al.* Single-cell mass cytometry of differential immune and drug responses across a human hematopoietic continuum. *Science* **332**, 687-696, doi:10.1126/science.1198704 (2011).
- 3 Van Gassen, S. *et al.* FlowSOM: Using self-organizing maps for visualization and interpretation of cytometry data. *Cytometry. Part A : the journal of the International Society for Analytical Cytology* **87**, 636-645, doi:10.1002/cyto.a.22625 (2015).
- 4 Qiu, P. *et al.* Extracting a cellular hierarchy from high-dimensional cytometry data with SPADE. *Nature biotechnology* **29**, 886-891, doi:10.1038/nbt.1991 (2011).
- 5 Jorgensen, C. C., Pitter, F. T., Kehlet, H., Lundbeck Foundation Center for Fast-track, H. & Knee Replacement Collaborative, G. Safety aspects of preoperative high-dose glucocorticoid in primary total knee replacement. *British journal of anaesthesia* **119**, 267-275, doi:10.1093/bja/aeq190 (2017).
- 6 Milush, J. M. *et al.* Functionally distinct subsets of human NK cells and monocyte/DC-like cells identified by coexpression of CD56, CD7, and CD4. *Blood* **114**, 4823-4831, doi:10.1182/blood-2009-04-216374 (2009).
- 7 Fowler, A. J., Abbott, T. E. F., Prowle, J. & Pearse, R. M. Age of patients undergoing surgery. *The British journal of surgery* **106**, 1012-1018, doi:10.1002/bjs.11148 (2019).
- 8 Nikolich-Zugich, J. The twilight of immunity: emerging concepts in aging of the immune system. *Nature immunology* **19**, 10-19, doi:10.1038/s41590-017-0006-x (2018).
- 9 Goronzy, J. J. & Weyand, C. M. Mechanisms underlying T cell ageing. *Nature reviews. Immunology* **19**, 573-583, doi:10.1038/s41577-019-0180-1 (2019).
- 10 Gaudilliere, B. *et al.* Clinical recovery from surgery correlates with single-cell immune signatures. *Science translational medicine* **6**, 255ra131, doi:10.1126/scitranslmed.3009701 (2014).
- 11 Lunn, T. H. *et al.* Effect of high-dose preoperative methylprednisolone on pain and recovery after total knee arthroplasty: a randomized, placebo-controlled trial. *British journal of anaesthesia* **106**, 230-238, doi:10.1093/bja/aeq333 (2011).
- 12 Miyagawa, Y. *et al.* Methylprednisolone reduces postoperative nausea in total knee and hip arthroplasty. *J Clin Pharm Ther* **35**, 679-684, doi:10.1111/j.1365-2710.2009.01141.x (2010).
- 13 Kehlet, H. & Lindberg-Larsen, V. High-dose glucocorticoid before hip and knee arthroplasty: To use or not to use-that's the question. *Acta orthopaedica* **89**, 477-479, doi:10.1080/17453674.2018.1475177 (2018).
- 14 Kehlet, H. & Joshi, G. P. The systematic review/meta-analysis epidemic: a tale of glucocorticoid therapy in total knee arthroplasty. *Anaesthesia*, doi:10.1111/anae.14946 (2019).
- 15 Weikum, E. R., Knuesel, M. T., Ortlund, E. A. & Yamamoto, K. R. Glucocorticoid receptor control of transcription: precision and plasticity via allosteric. *Nature reviews. Molecular cell biology* **18**, 159-174, doi:10.1038/nrm.2016.152 (2017).
- 16 Luecke, H. F. & Yamamoto, K. R. The glucocorticoid receptor blocks P-TEFb recruitment by NFkappaB to effect promoter-specific transcriptional repression. *Genes & development* **19**, 1116-1127, doi:10.1101/gad.1297105 (2005).

REVIEWERS' COMMENTS:

Reviewer #1 (Remarks to the Author):

The authors answered my questions and concerns with clear descriptions. The manuscript was revised to improve on the presentation of data and to clarify the weaknesses that were noted in the first submission. In particular, this reviewer was convinced by the validation data that was presented for their signaling studies by CyTOF and by the discussion of their analytical workflow.

Reviewer #2 (Remarks to the Author):

The authors satisfactorily responded to my concerns. They now discuss the limitations caused by the restriction of the study to elder individuals and the single dosing of MP, and also provide some confirmatory data concerning my former concern. Although Suppl. Fig. 4 is still very small, it is now readable. In this respect, my only suggestion would be to enlarge the asterisks indicating statistical significance since they are easily confounded with data points. The statement concerning the inhibition of NF-kB activity by GCs has been revised. I accept that providing additional data on functional features of immune cells would probably go beyond the scope of the current manuscript. I now recommend publication of this work.

Holger Reichardt

REVIEWERS' COMMENTS:

Reviewer #1 (Remarks to the Author):

The authors answered my questions and concerns with clear descriptions. The manuscript was revised to improve on the presentation of data and to clarify the weaknesses that were noted in the first submission. In particular, this reviewer was convinced by the validation data that was presented for their signaling studies by CyTOF and by the discussion of their analytical workflow.

We appreciate the reviewer's careful and thorough analysis and comments.

Reviewer #2 (Remarks to the Author):

The authors satisfactorily responded to my concerns. They now discuss the limitations caused by the restriction of the study to elder individuals and the single dosing of MP, and also provide some confirmatory data concerning my former concern. Although Suppl. Fig. 4 is still very small, it is now readable. In this respect, my only suggestion would be to enlarge the asterisks indicating statistical significance since they are easily confounded with data points. The statement concerning the inhibition of NF- κ B activity by GCs has been revised. I accept that providing additional data on functional features of immune cells would probably go beyond the scope of the current manuscript. I now recommend publication of this work.

We appreciate the reviewer's concerns and careful critique. In response, Supplementary Figure 4 has been subdivided into 12 separate figures in the interest of legibility and clarity.